# Chinese Rural Kindergarten Teachers’ Work–Family Conflict and Their Turnover Intention: The Role of Emotional Exhaustion and Professional Identity

**DOI:** 10.3390/bs14070597

**Published:** 2024-07-14

**Authors:** Yingjie Wang, Qianqian Xia, Huilan Yue, Wei Teng

**Affiliations:** 1School of Teacher Education, Huzhou University, Huzhou 313000, China; 03066@zjhu.edu.cn (Y.W.); 2022384537@stu.zjhu.edu.cn (Q.X.); yuehuilan@zjhu.edu.cn (H.Y.); 2School of Early Childhood Education, Shanghai Normal University Tianhua College, Shanghai 201815, China

**Keywords:** work–family conflict, turnover intention, emotional exhaustion, professional identity, rural kindergarten teacher

## Abstract

The loss of rural kindergarten teachers has become a common social concern in China, which is of great importance to the development of preschool education. This study conducted a survey of 2944 kindergarten teachers in mainland China, to explore the relationship between work–family conflict and turnover intention, the mediating effect of emotional exhaustion, and the moderating effect of professional identity. The study used the work–family conflict questionnaire, the emotional exhaustion scale, the turnover intention questionnaire, and the professional identity questionnaire. The results showed that (1) work–family conflict significantly predicted turnover intention; (2) emotional exhaustion played a mediating role between work–family conflict and turnover intention; and (3) professional identity moderated the latter half path of the mediation model, that is, strong professional identity alleviated the indirect predicting effect of work–family conflict on turnover intention through emotional exhaustion. The results clarified the influencing mechanism of work–family conflict on turnover intention, which could help improve rural preschool teachers’ positive emotions and reducing turnover.

## 1. Introduction

The Action Plan for the Revitalization of Teacher Education in China (2018–2022) proposes to strengthen the construction of rural preschool education and strive to improve the quality of rural kindergarten teachers [1]. The development of rural preschool education is of great significance to rural revitalization. Ensuring the quantity and quality of rural kindergarten teachers is the basis of revitalizing rural preschool education. However, in rural areas of China, the loss of kindergarten teachers is still a serious problem, and rural kindergartens face increasing difficulties retaining their teachers [2]. Due to the limited working and living conditions in rural areas, kindergarten teachers face significant professional pressure, such as limited income and heavy workloads, which challenge their stability and professional development [3,4,5]. A survey of 14,392 kindergarten teachers in six provinces of China found that 38.3% of kindergarten teachers have turnover intention, and this tendency is very prominent [6]. The turnover intention of rural kindergarten teachers is more obvious. High turnover rates affect teachers’ stability and hinder the quality of early childhood education [7,8]. In addition, a high turnover rate can damage the stable relationship between teachers and children, increase emotional pressure on children, and affect children’s physical and mental development [9]. Therefore, it is critical to explore the factors influencing the turnover intention of rural kindergarten teachers and implement effective measures to improve the quality of preschool education in rural areas. According to the Work–Family Interface Model, individuals will experience high levels of work–family conflict when work and family responsibilities interfere with each other [10]. Rural kindergarten teachers, who are predominantly women, face the challenge of balancing work and family responsibilities [11]. Consequently, they often experience greater pressure in both domains and are more likely to experience conflicts between work and family [12]. Therefore, examining the relationship between work–family conflict and turnover intention and the influence mechanism of rural kindergarten teachers can provide relevant recommendations for policymakers and kindergarten managers to alleviate teachers’ work–family conflict and reduce their turnover intention. It is also important to improve the stability of this group, promote early childhood development, and enhance the quality of rural preschool education.

### 1.1. Work–Family Conflict and Turnover Intention

Work and family are the two most important fields of individual life, and their conflict affects individual job burnout and mental health [13]. Work–family conflict refers to the difficulty for individuals to balance the different emotional and behavioral needs of work and family, resulting in role conflict [10,14], which include work leading to family conflict and family leading to work conflict [15]. Most kindergarten teachers are women who are influenced by traditional Chinese culture; consequently, they take on more family responsibilities, such as housework and childcare, while also handling heavy workloads and coping with various pressures at work [16,17,18]. Rural kindergarten teachers often face a higher degree of work–family conflict due to poor living and working conditions, which may result in increased pressure from both the family and work domains [19,20]. Moreover, due to the conflict between the shortage of teachers and the increase in non-teaching responsibilities in rural schools, teachers have to undertake a large number of teaching and non-teaching tasks [21]. In addition, in the context of China’s urbanization, teachers who return to the city after work have become the main group of rural teachers [22]. Due to heavy work tasks and frequent back and forth between school and family, rural teachers have no time to take care of their families, and the conflicts between work and family may become more intense.

According to the Scarcity Hypothesis, work–family conflict is the result of competition between individuals’ limited resources [23]. When coping with the demands of work and family, individuals consume resources, and the increase in resource consumption in one domain inevitably leads to a decrease in available resources in the other domains. Consequently, individuals adopt defensive behaviors, including an increased willingness to leave the organization and quit their jobs, to protect their resources [24,25]. Turnover intention refers to the conscious and deliberate willfulness to leave the organization, which is the most direct antecedent variable and best predictor of turnover behavior [26]. Empirical studies have found that work–family conflict could predict employees’ turnover intention [27,28]. Yildiz et al. [29] found that work–family conflict decreased nurses’ positive resources, such as mental and physical health, leading them to leave their jobs to protect their resources. Claflin et al. [27] found that career and technical education teachers often neglected their families due to their work responsibilities, which increased their psychological pressure, decreased their job satisfaction, and led to higher levels of turnover intention. Although previous studies have explored the relationship between work–family conflict and work outcomes, studies focusing on rural kindergarten teachers are scarce. Based on previous research, we hypothesized that work–family conflict among kindergarten teachers is positively associated with their turnover intention.

### 1.2. The Mediating Role of Emotional Exhaustion

According to the Job Demands–Resources model, the factors that affect work outcomes are mainly divided into job resources and demands. Job demands are aspects of work that require continuous physical, energetic, and emotional effort and can lead to energy consumption. With the depletion of energy, individual stress increases, and physical and mental fatigue increasingly intensifies, ultimately leading to emotional burnout [30]. Work–family conflict is often classified as a type of job demand, which can consume time, energy, and emotional resources to fulfill family and work responsibilities [31,32]. Consequently, it triggers a series of stress reactions, such as emotional exhaustion, ultimately leading to negative work outcomes, such as absenteeism or turnover intention [33]. Emotional exhaustion refers to a fatigue state of physical and emotional depletion, with emotional resources becoming drained [34]. In the state of emotional exhaustion, individuals experience negative emotions, feel tired of engaging in their work, and lack dedication and commitment to work [35]. Many empirical studies have shown that work–family conflict is an important predictor of emotional exhaustion [36,37]. For instance, Tian et al. [38] found that rural school teachers’ work–family conflict was positively related to their negative working emotion. Liu et al. [39] found primary and secondary school teachers’ work–family conflict can predicted their emotional exhaustion. Zhou et al. [40] found that excessive job demands caused preschool teachers to focus their personal resources on work, leaving fewer resources available to meet the demands of their families; combined with insufficient social support, the accumulation of work and family conflict resulted in high levels of emotional exhaustion.

In addition, a previous study found that individuals who suffer from emotional exhaustion may resort to negative organizational behaviors, such as resigning from their jobs [41]. Some empirical studies have shown that emotional exhaustion reduces job satisfaction, impedes in-role and out-of-role performance in the work and family domains, and leads to high levels of turnover intention [42,43]. A study on female employees of township banks showed that women were often burdened with the primary responsibility of managing family affairs, leading to work–family conflicts that could result in fatigue and burnout, and ultimately prompting a desire to leave their job [44,45]. Similarly, a study on hotel employees found that, due to the particularity of their profession, they were required to engage in a significant amount of emotional labor [46]. The lack of adequate compensation for their psychological capital was likely to decrease their work performance and increase their intention to leave [47,48]. Moreover, Ding and Lyu [49] found that special education teachers who experienced emotional exhaustion were more likely to adopt coping strategies of avoidance or withdrawal, such as resignation, to reduce the psychological cost of emotional exhaustion. Burić et al. [50] also found that teachers’ work–family conflict could predict their professional commitment through emotional exhaustion. Su and Jiang [51] found that university teachers’ work–family conflict can predict their job satisfaction through job burnout (include emotional exhaustion). The Job Demands–Resources model and previous empirical studies prove that teachers’ work–family conflict may influence their work outcomes through their working emotion. Furthermore, the variables of work–family conflict, emotional exhaustion, and turnover intention are related to each other. So, we hypothesized that the emotional exhaustion of rural kindergarten teachers plays a mediating role in the relationship between work–family conflict and turnover intention.

### 1.3. The Moderating Role of Professional Identity

According to the Risk-Buffering Hypothesis, from the perspective of individual career development, risk factors do not necessarily lead to individual maladjustment, and positive individual characteristics can buffer the negative effects of risk factors [52,53]. Although many empirical studies have shown that individuals who experience emotional exhaustion are more likely to have a higher level of turnover intention [42,49], positive individual characteristics may buffer this negative effect. A previous study found that professional identity buffered the negative effect of job burnout on turnover intention among preschool teachers [54]. Professional identity refers to an individual’s positive attitude and strong sense of commitment to a certain occupation, which is reflected in their desire to maintain their occupation and feeling a degree of attachment to it [55]. Teachers’ professional identity is an important indicator measuring the quality of their work and helps improve their job satisfaction, enthusiasm, and commitment [56]. According to the Price–Mueller model, an individual’s turnover intention is affected by environmental, individual, and organizational factors [57]. Emotions are one of the most important individual factors affecting teachers’ turnover intention. Individuals with positive emotions have a strong sense of career belief and positively identify with their work value. Individuals with high levels of professional identity perceive the favorable aspects of work to develop more psychological resources, leading to higher job satisfaction and organizational commitment, which helps buffer against emotional exhaustion [5,58,59] and reduce turnover intention [60,61]. In contrast, individuals with low levels of professional identity are more susceptible to experiencing negative emotions in the workplace and may suffer from physical and psychological exhaustion, which increases their turnover intention [62,63]. Empirical studies have confirmed that professional identity reduces job burnout and turnover intention [64,65]. A study on nurses found that high levels of professional identity reduced the negative consequences of job burnout, such as low job satisfaction and work efficiency [64]. Ding [54] found that high professional identity alleviated the influence of job burnout on turnover intention among college counselors. Therefore, professional identity plays an important role in reducing the negative impact of emotional exhaustion on turnover intention. Based on this, we hypothesized that professional identity has a moderating effect on the relationship between emotional exhaustion and turnover intention of rural kindergarten teachers.

### 1.4. The Present Study

In the context of China’s urbanization, many rural school teachers rush between rural areas and cities for work and life every day, and the female-dominated kindergarten teachers also have to shoulder the responsibility of taking care of their families. So, work–family conflict and turnover intention among rural kindergarten teachers are more obvious, and more attention needs to be paid to them. According to the theoretical framework of the Scarcity Hypothesis, the Job Demands–Resources model, and the Risk-Buffering Hypothesis, as well as the empirical studies, work–family conflict, emotional exhaustion, and professional identity are important factors affecting the turnover intention of kindergarten teachers. Furthermore, teachers’ work–family conflict can affect their work outcomes through working emotions. Many studies mentioned above have proved that professional identity was usually used as a moderating agent to mitigate the negative impact of adverse organizational environment factors. However, most of the existing studies mainly focused on primary and middle school teachers, and fewer studies focused on kindergarten teachers, especially rural kindergarten teachers. In China, the number of kindergarten teachers in rural areas has increased rapidly, but the turnover rate is also very large. It is of great significance to explore the mechanism of rural kindergarten teachers’ turnover intention from the perspective of work–family conflict. In addition, among the existing studies, few of them included all these variables in one model to examine their combined effect on teachers’ turnover intention. Therefore, this study aimed to explore the effect of work–family conflict on turnover intention among rural kindergarten teachers, the mediating effect of emotional exhaustion, and the moderating effect of professional identity. And then we propose the following research hypotheses: H1. Work–family conflict is positively related to turnover intention. H2. Emotional exhaustion mediates the relationship between work–family conflict and turnover intention. H3. Professional identity moderates the latter half path of the mediation model. The proposed hypothetical model is illustrated in Figure 1.

## 2. Materials and Methods

### 2.1. Participants

Random sampling was used and this study recruited 3270 kindergarten teachers from 120 public kindergartens located in rural areas of East Mainland China in 2022. After data checking, 326 questionnaires were excluded because of incomplete data. Eventually, a total of 2944 rural kindergarten teachers were included during the following analysis, with 2914 females and 30 males. The mean teaching experience was 9.66 years, *SD* = 7.23 years. Additionally, 40 teachers possessed a high school diploma, 1013 teachers had a college degree, and 1891 teachers had a Bachelor’s degree.

### 2.2. Procedure

This study used an online questionnaire platform called “Wen juan xing”, and the questionnaires were filled out by rural kindergarten teachers in the east of China. The ethical approval of this study was provided by the Institutional Review Board of the researchers’ university. We contacted the principals of each kindergarten, and with the help of the principals, the online questionnaire link of data collection was sent to each teacher and they were informed of the precautions for filling in the questionnaires. All the teachers participated voluntarily. The informed consent for this study was obtained prior to formal data collection, during which we provided teachers with a comprehensive explanation of the questionnaire’s details and requirements. All participants needed to fill in the demographic information and the questionnaires of teachers’ work–family conflict, emotional exhaustion, turnover intention, and professional identity. After completing all items, all participants could receive a small gift as a reward.

### 2.3. Measures

#### 2.3.1. Work–Family Conflict

Work–family conflict was measured using the Work–Family Conflict Scale [66]. This scale includes 10 items and encompasses two dimensions: work interferes with family (e.g., “My job makes it difficult for me to fulfill my family responsibilities”) and family interferes with work (e.g., “I have to put off my work due to family reasons”). Kindergarten teachers rated items on a five-point Likert scale ranging from 1 (completely disagree) to 5 (completely agree). Higher scores indicated a greater degree of work–family conflict experienced by teachers. Cronbach’s α coefficient of the scale in this study was 0.89.

#### 2.3.2. Emotional Exhaustion

Emotional exhaustion was measured using the teachers’ Job Burnout Scale [35], which consists of three subscales. We used the Emotional Exhaustion subscale (9 items, e.g., “I feel a loss of enthusiasm for teaching”). The measurement was evaluated utilizing a seven-point Likert scale, ranging from 1 (never) to 7 (always), with higher scores corresponding to higher levels of emotional exhaustion. Cronbach’s α coefficient of the emotional exhaustion subscale in the present study was 0.94.

#### 2.3.3. Turnover Intention

Turnover intention was measured using the Turnover Intention Scale [67], which consists of four items (e.g., “If it is possible, I would like to quit this job”). Responses were rated on a seven-point Likert scale (1 = never; 7 = always), with higher score indicating higher levels of turnover intention. Cronbach’s α coefficient of the scale in this study was 0.87.

#### 2.3.4. Professional Identity

The measurement of the teacher’s professional identity was conducted using the Professional Identity Scale [68]. The scale comprises a total of 18 items (e.g., I am proud of myself to be a preschool teacher). The participants were asked to rate their responses using a five-point Likert scale (1 = very inconsistent; 5 = very consistent), with higher scores corresponding to higher levels of professional identity. Cronbach’s α coefficient of the scale in this study was 0.92.

### 2.4. Demographic Information

In this study, teachers’ demographic information was also included. According to previous studies, some teachers’ demographic information was related to the main study variables, such as work–family conflict, emotional exhaustion, and turnover intention [36,49]. Therefore, during the subsequent analysis, we controlled the demographic information of teachers’ age, gender, teaching experience, educational background, and income.

### 2.5. Data Analysis

All statistical analyses were performed utilizing SPSS 25.0. First, the correlation analysis was conducted for all variables. Second, the SPSS PROCESS software (Version 3.3) was utilized to perform the test for assessing the mediation analysis and the moderated mediation effect [69]. Third, we used a bootstrapping method that included 5000 resamples to evaluate the unconditional indirect effects. When the 95% confidence intervals (95% CI) did not contain zero, we considered the effect to be statistically significant.

## 3. Results

### 3.1. Descriptive and Correlational Analysis

Table 1 presents the means, SDS, and correlations for demographic information and main study variables (work–family conflict, emotional exhaustion, turnover intention, and professional identity). The results revealed a significant positive correlation between work–family conflict and emotional exhaustion (*r* = 0.60, *p* < 0.001) as well as turnover intention (*r* = 0.49, *p* < 0.001). Additionally, work–family conflict exhibited a significant negative correlation with professional identity (*r* = −0.28, *p* < 0.001). Emotional exhaustion was significantly positively correlated with turnover intention (*r* = 0.67, *p* < 0.001) and significantly negatively correlated with professional identity (*r* = −0.38, *p* < 0.001). Turnover intention was significantly negatively correlated with professional identity (*r* = −0.34, *p* < 0.001). The results of the correlational analysis indicated that the demographic information was significantly correlated to the study variables, so, during the subsequent analysis, the demographic information were controlled. The results also showed the four study variables were significantly correlated to each other, which meets the basic conditions of the mediating and moderating effect analysis. So, we subsequently conducted the mediation and moderated mediation analysis.

### 3.2. Mediation Analysis

We used Model 4 of the SPSS PROCESS to examine the mediating effect of emotional exhaustion. The demographic characteristics were included as control variables in order to account for their potential influence. According to Figure 2 and Table 2, work–family conflict was positively associated with turnover intention (*β* = 0.49, *p* < 0.001). After the inclusion of emotional exhaustion to the model, work–family conflict could also positively predict turnover intention (*β* = 0.13, *p* < 0.001). In addition, work–family conflict significantly predicted emotional exhaustion (*β* = 0.59, *p* < 0.001), and emotional exhaustion positively predicted turnover intention (*β* = 0.60, *p* < 0.001). To investigate the mediating effect, we employed the bias-corrected bootstrap technique and performed 5000 bootstrap samples using PROCESS. The results revealed a significant indirect effect with a value of 0.35 and a 95% confidence interval ranging from −0.10 to −0.07, which excluded zero. So, the mediating effect of emotional exhaustion between work–family conflict and turnover intention was significant. The results indicated that work–family conflict can lead to teachers’ turnover intention via their emotional exhaustion during work.

### 3.3. Moderated Mediation Analysis

We examined the moderating effect of professional identity in the mediation model by employing Model 14 of PROCESS. The results presented in Table 3 showed that the interaction effect of teachers’ emotional exhaustion and professional identity on their turnover intention was significant (*β*= −0.04, t = −3.44, *p* < 0.01), which indicated that the moderating effect of professional identity on the relationship between emotional exhaustion and turnover intention was significant. That is, the influence of teachers’ emotional exhaustion on their turnover intention was affected by the level of teachers’ professional identity.

In addition, we conducted a simple slope analysis (Figure 3) to explore the moderating mechanism of professional identity. We divided rural kindergarten teachers into a high level (+1SD) and a low level (−1SD) group based on their professional identity scores. The predictive effect of emotional exhaustion on turnover intention was lower for teachers with a high level of professional identity (*β* = 0.53, *t* = 25.22, *p* < 0.001) than those teachers with a lower level of professional identity (*β* = 0.62, *t* = 29.97, *p* < 0.001). The result indicated that when rural kindergarten teachers were at a higher level of professional identity, the negative affect of work–family conflict on their turnover intention through emotional exhaustion is lower than in the case of teachers with a low level of professional identity. So, the result proved that professional identity is a protective factor of teachers’ turnover intention, which can buffer the negative affect brought on by teachers’ work–family conflict and emotional exhaustion.

## 4. Discussion

This study analyzed the relationship between work–family conflict and turnover intention among rural kindergarten teachers, and found the mediating effect of emotional exhaustion and the moderating effect of professional identity. With the development of society, more and more women enter the workplace; most of them are faced with the problem of balancing work and family, and kindergarten teachers are more prominent. This study revealed the influence mechanism of rural kindergarten teachers’ work–family conflict on their turnover intention against the background of the new era, and provided the empirical basis for the professional development of rural kindergarten teachers and the practice of stabilizing kindergarten teachers in rural areas for the policymakers.

### 4.1. Relationship between Work–Family Conflict and Turnover Intention

This study found that work–family conflict was positively related to turnover intention, which was consistent with previous studies [8,70]. In addition, the result was also consistent with the Scarcity Hypothesis [71]. The Scarcity Hypothesis posits that individuals possess a limited amount of time and energy, and overcommitment to one role can reduce the resources available for another, while simultaneous participation in multiple roles may deplete resources and result in role conflict [72,73]. When individuals experience work–family conflict, they may try to resolve it by quitting their jobs [23,74]. In traditional Chinese culture, women are expected to shoulder more responsibility for taking care of their family; however, with the development of society, an increasing number of women enter the workforce and are responsible for many work tasks [75]. Therefore, women face great challenges in balancing family obligations and work responsibilities [17,76]. Rural kindergarten teachers, who are predominantly women, not only shoulder more family responsibilities, such as housework and childcare, but also engage in heavy work tasks [77,78]. Consequently, they are more likely to experience conflicts resulting from the improper management of the relationship between work and family [12,76]. Kindergarten teachers who experience higher levels of work–family conflict may encounter greater work pressure, affecting their investment in either work or family. Consequently, they may be more inclined to quit their job to avoid these negative experiences [70,79].

### 4.2. Mediating Role of Emotional Exhaustion

This study found that emotional exhaustion played a mediating role in the relationship between work–family conflict and turnover intention. Previous studies have found that work–family conflict triggers employees’ negative emotions, leading to negative work outcomes, such as low job satisfaction, high turnover intention, and low work efficiency [32,33,36]. This is in line with the results of the present study. The results are also consistent with the Job Demands–Resources Model, which suggests that high job demands deplete individuals’ energy and resources, leading to negative outcomes, such as emotional exhaustion, and consequently reducing work commitment and increasing turnover intention [31,80]. Kindergarten teachers are often faced with heavy workloads, which include teaching, classroom management, and parenting [81,82]. Teachers in rural areas face greater pressure from children’s parents and social demands regarding their roles [20,40,75]. When excessive work demands intrude into the family domain and cause work–family conflict, teachers will experience higher levels of stress that reduce their effectiveness in both domains. To cope with negative emotions, teachers further deplete existing emotional resources and suffer from emotional exhaustion [83,84]. Emotional exhaustion can result in decreased work efficiency and reduced organizational commitment among teachers, so they will protect themselves from further harm by showing negative organizational behaviors, such as resignation, to reduce the psychological cost of emotional exhaustion [45,85]. On the contrary, kindergarten teachers who are highly emotional laborers, must invest more emotions into their profession and children, and those who experience less emotional exhaustion are able to devote more energy to their work, thus alleviating their feelings of alienation towards their work and decreasing their intention to leave [86].

### 4.3. Moderating Effect of Professional Identity

This study found that professional identity moderated the association between emotional exhaustion and turnover intention in the mediating model. In other words, compared with the teachers with a lower level of professional identity, those with a higher level of professional identity were better able to maintain a stronger sense of professional belonging despite experiencing emotional exhaustion. A high level of professional identity reduced the negative impact of work–family conflict on turnover intention through emotional exhaustion. According to the Price–Mueller model, individuals with positive professional emotions have high levels of professional identity and they can effectively deal with negative emotions, which in turn leads to confidence and enthusiasm for their work and reduces negative organizational behaviors caused by emotional exhaustion [54,57]. Teachers with high levels of professional identity usually have a strong sense of professional belonging and trust in their careers even when faced with complicated work [87,88]. Teachers with low levels of professional identity lack enthusiasm and responsibility towards their work, often lose confidence in their careers due to insufficient support and intrinsic motivation when confronted with emotional problems, and may eventually choose to leave their jobs [54,61]. In addition, teachers with a high level of professional identity actively engage in education and teaching, which alleviates the negative impact of unfavorable working environment and conditions on job satisfaction and burnout, thereby improving job stability [58,89,90].

### 4.4. Implications and Limitations

Based on the survey of 2944 rural kindergarten teachers in China, this study identified the mediating effect of emotional exhaustion and moderating effect of professional identity on the relationship between teachers’ work–family conflict and their turnover intention. These results have significant theoretical implications. First, the results clarified the influence mechanism of work–family conflict on turnover intention of rural kindergarten teachers and the protective role of professional identity. Moreover, the results also highlighted the negative effect of work–family conflict and the buffering effect of professional identity, which expanded the interpretation scope of existing theoretical models, such as the Job Demands–Resources model, the Scarcity Hypothesis, and the Risk-Buffering Hypothesis.

The results of the study also have important practical implications. Firstly, the results enlightened the kindergarten managers’ need to enhance the professional identity of rural kindergarten teachers in various ways, such as strengthening their professional identity education, improving the treatment of teachers, providing more humanistic care for teachers, and reducing their professional pressure. Secondly, the policymakers also should pay more attention to the negative effect brought on by the work–family conflict of rural kindergarten teachers. Some measures should be taken to ease their work–family conflict and emotional exhaustion, for example, providing convenience for rural kindergarten teachers to commute and reducing their time cost of commuting; increasing the financial subsidies for kindergarten teachers in rural areas; and providing specialized mental health training. Thirdly, the teachers should also balance family life and work tasks well, and regulate their emotional state.

This study had several limitations. First, a cross-sectional design was used to collect data; therefore, causal relationships among the variables could not be examined. Future studies should implement a longitudinal design to explore the associations among the study variables. Second, the data used in this study were from one province, which is a relatively developed province in terms of education. Therefore, the generalizability of the findings may be limited, and subsequent sampling from broader areas (especially the west provinces in China) is necessary. Teachers’ work–family conflict and their turnover intention may be influenced by family factors (e.g., family support), but this study only focused on the organizational and environmental factors. So, future studies can incorporate family factors into the model to investigate the potential mechanisms.

## 5. Conclusions

This study revealed significant findings regarding rural kindergarten teachers. Work–family conflict significantly and directly predicted turnover intention. Emotional exhaustion mediated the relationship between work–family conflict and turnover intention. Professional identity moderated the latter half path of the mediation model; that is, a strong professional identity buffered the predicting effect of work–family conflict on turnover intention through emotional exhaustion. These findings provide a new perspective for expanding the theory of work–family conflict and teachers’ turnover intention. Furthermore, the results provide important insights for policymakers, kindergarten managers, and teachers themselves to take positive measures to enhance professional identity, and ease work–family conflict and emotional exhaustion.

## Figures and Tables

**Figure 1 behavsci-14-00597-f001:**
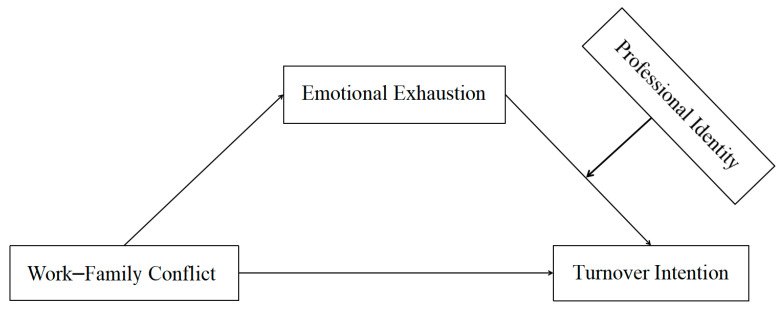
Hypothesis model.

**Figure 2 behavsci-14-00597-f002:**
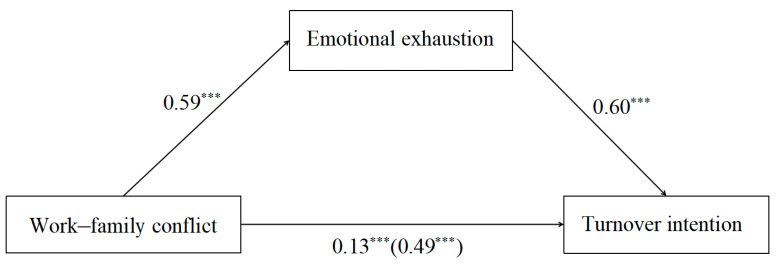
The mediating effect of emotional exhaustion. Note: *** *p* < 0.001.

**Figure 3 behavsci-14-00597-f003:**
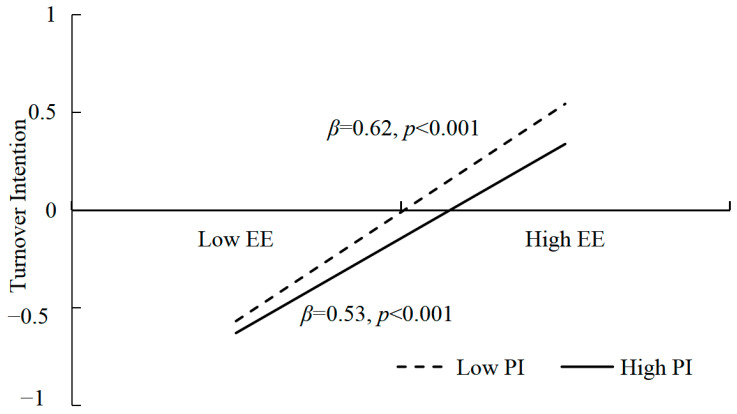
The moderating effect of professional identity between emotional exhaustion and turnover intention. Note: PI = professional identity; EE = emotional exhaustion.

**Table 1 behavsci-14-00597-t001:** Correlations and descriptive statistics among the main variables.

	1	2	3	4	5	6	7	8	9
1. Gender	-								
2. Age	0.02	-							
3. Teaching experience	0.02	0.84 ***	-						
4. Educational background	0.02	0.06 *	0.10 ***	-					
5. Income	−0.05 *	0.09 ***	0.16 ***	0.33 ***	-				
6. Work–family conflict	−0.01	−0.07 ***	−0.05 *	0.04 *	0.02	-			
7. Emotional exhaustion	0.02	0.07 ***	−0.03	0.07 ***	0.03	0.60 ***	-		
8. Turnover intention	0.03	−0.09 ***	−0.05 *	−0.04 *	−0.16 ***	0.49 ***	0.67 ***	-	
9. Professional identity	0.01	0.06 **	0.06 **	0.06 ***	0.08 ***	−0.28 ***	−0.38 ***	−0.34 ***	-
*M*	-	32.08	9.66	-	-	2.43	3.11	2.91	4.32
*SD*	-	7.10	7.23	-	-	0.78	1.22	1.33	0.57

Note: * *p* < 0.05, ** *p* < 0.01, *** *p* < 0.001.

**Table 2 behavsci-14-00597-t002:** Mediation model results.

Dependent Variables	Independent Variables	*R*	*R* ^2^	*F*	*β*	*t*	*95% CI*
Turnover intention		0.52	0.27	183.44 ***			
	Gender				0.25	1.71	[−0.04, 0.54]
	Age				−0.02	−4.50 ***	[−0.03, −0.01]
	Teaching experience				0.01	3.59 ***	[0.01, 0.02]
	Educational background				−0.01	−0.36	[−0.08, 0.05]
	Income				−0.18	−9.98 ***	[−0.22, −0.15]
	Work–family conflict				0.49	30.80 ***	[0.46, 0.52]
Emotional exhaustion		0.60	0.37	281.03 ***			
	Gender				0.22	1.58	[−0.05, 0.49]
	Age				−0.02	−4.05 ***	[−0.02, −0.01]
	Teaching experience				0.01	3.14 **	[0.01, 0.02]
	Educational background				0.07	2.38 **	[0.01, 0.13]
	Income				0.01	0.30	[−0.03, 0.04]
	Work–family conflict				0.59	40.19 ***	[0.56, 0.62]
Turnover intention		0.71	0.50	422.84 ***			
	Gender				0.13	0.10	[−0.12, 0.36]
	Age				−0.01	−2.68 **	[−0.02, −0.01]
	Teaching experience				0.01	2.21 *	[0.001, 0.01]
	Educational background				−0.06	−2.04 *	[−0.11, −0.01]
	Income				−0.19	12.26 ***	[−0.21, −0.16]
	Emotional exhaustion				0.60	36.77 ***	[0.57, 0.63]
	Work–family conflict				0.13	7.97 ***	[0.10, 0.16]

Note: * *p* < 0.05, ** *p* < 0.01, *** *p* < 0.001.

**Table 3 behavsci-14-00597-t003:** Results of the moderated mediation model.

Dependent Variables	Independent Variables	*R*	*R* ^2^	*F*	*β*	*t*	*95% CI*
Emotional exhaustion		0.60	0.37	281.03 ***			
	Gender				0.22	1.58	[−0.05, 0.49]
	Age				−0.02	−4.05 ***	[−0.02, −0.01]
	Teaching experience				0.01	3.14 **	[0.00, 0.01]
	Educational background				0.07	2.38 *	[0.01, 0.13]
	Income				0.01	0.30	[−0.03, 0.04]
	Work–family conflict				0.59	40.19 ***	[0.56, 0.62]
Turnover intention		0.71	0.51	338.09 ***			
	Gender				0.14	1.15	[−0.10, −0.38]
	Age				−0.01	−2.63 **	[−0.02, −0.00]
	Teaching experience				0.01	2.34 *	[0.00, 0.01]
	Educational background				−0.05	−1.77	[−0.10, 0.01]
	Income				−0.18	−12.03 ***	[−0.21, −0.15]
	Work–family conflict				0.13	7.93 ***	[0.10, 0.16]
	Emotional exhaustion				0.57	33.92 ***	[0.54, 0.61]
	Professional identity				−0.07	−5.24 ***	[−0.10, −0.05]
	Emotional exhaustion × professional identity				−0.04	−3.44 **	[−0.07, −0.02]

Note: * *p* < 0.05, ** *p* < 0.01, *** *p* < 0.001.

## Data Availability

The data in the study is available from the corresponding author upon reasonable request.

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
