# Peer review of "Chinese Rural Kindergarten Teachers’ Work–Family Conflict and Their Turnover Intention: The Role of Emotional Exhaustion and Professional Identity"

_behavsci, 2024, doi:10.3390/bs14070597_

Round 1

Reviewer 1 Report

Comments and Suggestions for Authors

The manuscript is well structured and the theoretical frameworks and the hypotheses were well developed. The only weakness I can find is that I don't see any variables controlled. That limited the significance of the study. For instance, how does gender, age, family status, and education level explain the relationships proposed in the hypotheses model? That analysis can help us better understand the kindergarden teachers population and useful for providing meaningful implications to practice. 

Comments on the Quality of English Language

Clear. Just minor edits are needed. 

Author Response

Dear Reviewer:

Thank you for your letter and for the reviewers’ comments concerning our manuscript entitled “Chinese Rural Kindergarten Teachers’ Work–Family Conflict and Their Turnover Intention: The Role of Emotional Exhaustion and Professional Identity” (ID: behavsci-3034573). Those comments are all valuable and very helpful for revising and improving our paper, as well as the important guiding significance to our researches. We have studied comments carefully and have made correction which we hope meet with approval. Revised portion are highlighted with yellow in the paper. The main corrections in the paper and the responds to the reviewer's comments are as flowing:

Comments1: [The manuscript is well structured and the theoretical frameworks and the hypotheses were well developed. The only weakness I can find is that I don't see any variables controlled. That limited the significance of the study. For instance, how does gender, age, family status, and education level explain the relationships proposed in the hypotheses model? That analysis can help us better understand the kindergarten teachers population and useful for providing meaningful implications to practice.]

Response1: [Thank you for the suggestions and the problems point out. In the study, the demographic information(e.g., teachers’ age, gender, teaching experience, educational background, and income) of rural kindergarten teachers was controlled during the data analysis, the effects of demographic information on the outcome variables were presented at table 2 and table 3. We also added a section of “2.4 Demographic information” on page 7 to describe the control variables, which was listed bellow:

2.4. Demographic information

In this study, teachers’ demographic information was also included. According to previous studies, some teachers’ demographic information was related to the main study variables, such as work-family conflict, emotional exhaustion, and turnover intention [36,49]. Therefore, during the following analysis, we controlled the demographic information of teachers’ age, gender, teaching experience, educational background, and income.]

Reviewer 2 Report

Comments and Suggestions for Authors
  • The introduction section of the article does not clearly outline the common ground, complication, course of action, or contribution. These elements are essential to establish the context, highlight the research problem, propose the solution, and indicate the study's contribution to the field. Consider restructuring the introduction to explicitly address these components. Provide a comprehensive background that situates the research within the broader context, specify the problem being addressed, outline the research approach, and articulate the study's contributions to the field.

  • The plagiarism report indicates a 42% similarity index. This is significantly high and suggests potential issues with originality. It is crucial to review the manuscript for possible instances of improper citation or lack of originality. Ensure all sources are properly cited and consider rephrasing or rewriting sections that closely resemble other works. This will not only improve the manuscript's originality but also its credibility.

    • Is the content succinctly described and contextualized with respect to previous and present theoretical background and empirical research (if applicable) on the topic?

      • Must be improved: The current state of the content lacks sufficient contextualization with respect to existing literature. Strengthen the literature review to provide a comprehensive theoretical and empirical backdrop for the study.
    • Are the research design, questions, hypotheses, and methods clearly stated?

      • Must be improved: The research design, questions, hypotheses, and methods need to be articulated more clearly. This includes providing detailed descriptions of the methodologies employed and how they address the research questions and hypotheses.
    • Are the arguments and discussion of findings coherent, balanced, and compelling?

      • Can be improved: The discussion of findings needs to be more coherent and balanced. Ensure that the arguments are logically structured and supported by the data.
    • For empirical research, are the results clearly presented?

      • Can be improved: The presentation of results should be clearer. Use appropriate tables, figures, and statistical analyses to effectively communicate the findings.
    • Is the article adequately referenced?

      • Must be improved: The referencing needs enhancement. Ensure that all sources are cited correctly and that the reference list is complete and formatted according to the journal's guidelines.
    • Are the conclusions thoroughly supported by the results presented in the article or referenced in secondary literature?

      • Must be improved: The conclusions should be better supported by the results and relevant literature. Revisit the data to draw more substantiated conclusions.
  • Ratings:

    • Originality: Low
    • Contribution to Scholarship: Average
    • Quality of Structure and Clarity: Low
    • Logical Coherence/Strength of Argument/Academic Soundness: Low
    • Engagement with sources as well as recent scholarship: Average
    • Overall Merit: Low
  • Detailed Comments:

    1. Introduction:

      • Clearly define the common ground, complication, course of action, and contribution.
      • Provide a more detailed literature review that situates your research within the broader context of work-family conflict, emotional exhaustion, and professional identity.
    2. Methodology:

      • Ensure that the research design, questions, hypotheses, and methods are explicitly stated.
      • Describe the sampling method, data collection process, and statistical analyses in detail.
    3. Results:

      • Present the results more clearly, using tables and figures where appropriate.
      • Include more detailed explanations of the statistical analyses and their implications.
    4. Discussion:

      • Make the arguments more coherent and balanced.
      • Relate the findings to the existing literature and theoretical frameworks.
    5. Conclusion:

      • Ensure that the conclusions are thoroughly supported by the results.
      • Highlight the practical implications of the findings for policymakers and practitioners.
    6. References:

      • Review the reference list for completeness and accuracy.
      • Ensure all cited sources are relevant and up-to-date.

  •  

Author Response

Dear Reviewer:

Thank you for your letter and for the reviewers’ comments concerning our manuscript entitled “Chinese Rural Kindergarten Teachers’ Work–Family Conflict and Their Turnover Intention: The Role of Emotional Exhaustion and Professional Identity” (ID: behavsci-3034573). Those comments are all valuable and very helpful for revising and improving our paper, as well as the important guiding significance to our researches. We have studied comments carefully and have made correction which we hope meet with approval. Revised portion are highlighted with yellow in the paper. The main corrections in the paper and the responds to the reviewer's comments are as flowing:

Comment1: [The introduction section of the article does not clearly outline the common ground, complication, course of action, or contribution. These elements are essential to establish the context, highlight the research problem, propose the solution, and indicate the study's contribution to the field. Consider restructuring the introduction to explicitly address these components. Provide a comprehensive background that situates the research within the broader context, specify the problem being addressed, outline the research approach, and articulate the study's contributions to the field.]            

Response1: [Thank you for the suggestions and problems point out. According to the suggestions, we revised the part of introduction from a more comprehensive background under a broader context, and clearly presented their our research questions, and contributions to the field. Please see the yellow portion in the section of introduction.]

Comment2: [The plagiarism report indicates a 42% similarity index. This is significantly high and suggests potential issues with originality. It is crucial to review the manuscript for possible instances of improper citation or lack of originality. Ensure all sources are properly cited and consider rephrasing or rewriting sections that closely resemble other works. This will not only improve the manuscript's originality but also its credibility.]

Response2: [Thank you for the problems point out. According to the suggestions, we revised the whole article carefully, after finishing the revision, we used the software of “Turnitin” to test the similarity index, the exclude matches was set 1%, the result indicated a 12% similarity index, which meet the publication requirement.]

Comment3:

[Introduction:

Clearly define the common ground, complication, course of action, and contribution. Provide a more detailed literature review that situates your research within the broader context of work-family conflict, emotional exhaustion, and professional identity.]

Response3: [Thank you for the suggestions and the problems point out. According to the suggestions, we revised the whole introduction, we reviewed and cited the latest studies related to this article to further support the research hypothesis. In addition, we put our research problems under the context of “China's Urbanization”, “Rural Education Revitalization” , which introduced the effects of changes of society on rural kindergarten teachers’ work-family conflict, emotional exhaustion, turnover intention and professional identity. The revised part are highlighted with yellow in the section of Introduction. ]

Comment4:

[Methodology:

Ensure that the research design, questions, hypotheses, and methods are explicitly stated. Describe the sampling method, data collection process, and statistical analyses in detail.

Present the results more clearly, using tables and figures where appropriate.

Include more detailed explanations of the statistical analyses and their implications.]

Response4: [Thank you for the suggestions and the problems point out.

Firstly, we supplemented the statements of the research design, questions, and hypotheses in the part of 1.4 on page 5 to make it more clear. We also described the sampling method, data collection process, and statistical analyses in detail in the part of  “2. Materials and Methods” on page 6-7.

Secondly, we supplemented the statements of results, the results of mediation analysis and moderated mediation analysis were presented at table2-3 and figure2-3 in the section of “Results”on page8-11.

Thirdly, we added the detailed explanations of results of moderated mediation analysis in the section of 3.3 on the page of 9-10.]

Comment5:

[ Discussion:

 Conclusion: Ensure that the conclusions are thoroughly supported by the results.

Highlight the practical implications of the findings for policymakers and practitioners.

Response5: [Thank you for the suggestions and the problems point out.

Firstly, we checked the description of the conclusions to ensure that all the conclusions were supported by the results.

Secondly, according to the suggestions, we highlighted the practical implications of the findings for policymakers and practitioners in the section of 4.4 on page 12. The revised part was listed below:

The results of the study also enlighten the kindergarten managers need to enhance the professional identity of rural kindergarten teachers in various ways, such as strengthening their professional identity education, improving the treatment of teachers, providing more humanistic care for teachers, and reducing their professional pressure, etc. The policymakers also should pay more attention to the negative effect brought by work-family conflict of rural kindergarten teachers. Some measures should be taken to ease their work-family conflict and emotional exhaustion, for example, provide convenience for rural kindergarten teachers to commute and reduce their time cost of commuting; increase the financial subsidies for kindergarten teachers in rural areas; provide specialized mental health training, etc. The teachers should also balance family life and work tasks well, and regulate their emotional state.] 

Comment6: [References: Review the reference list for completeness and accuracy. Ensure all cited sources are relevant and up-to-date.]

Response6: [Thank you for the suggestions. We checked the citation and the references of the whole article to ensure all cited sources are relevant and up-to-date, and the reference list was complete and accurate.]

Reviewer 3 Report

Comments and Suggestions for Authors

The article demonstrates a high degree of systematicity and clarity, underpinned by a robust engagement with the extant research literature. The rationale for the study is articulated in the introduction with clarity and persuasiveness, effectively laying the groundwork for the ensuing research. The methodology employed is well-justified and rigorous, ensuring the reliability and validity of the findings. The presentation of results is methodical and transparent, allowing for a comprehensive understanding of the data.

One suggestion for improvement pertains to the discussion section. Specifically, it would be beneficial to enhance the creative dimension by integrating the various aspects currently addressed separately. A more cohesive synthesis of these elements would provide a richer, more nuanced interpretation of the findings. Additionally, while the article excels in systematicity and clarity, it would be advantageous to elucidate the implications of the findings within the broader context, thereby offering a more comprehensive perspective on the overall significance of the research.

Author Response

Dear Reviewer:

Thank you for your letter and for the reviewers’ comments concerning our manuscript entitled “Chinese Rural Kindergarten Teachers’ Work–Family Conflict and Their Turnover Intention: The Role of Emotional Exhaustion and Professional Identity” (ID: behavsci-3034573). Those comments are all valuable and very helpful for revising and improving our paper, as well as the important guiding significance to our researches. We have studied comments carefully and have made correction which we hope meet with approval. Revised portion are highlighted with yellow in the paper. The main corrections in the paper and the responds to the reviewer's comments are as flowing:

Comment1: [One suggestion for improvement pertains to the discussion section. Specifically, it would be beneficial to enhance the creative dimension by integrating the various aspects currently addressed separately. A more cohesive synthesis of these elements would provide a richer, more nuanced interpretation of the findings. Additionally, while the article excels in systematicity and clarity, it would be advantageous to elucidate the implications of the findings within the broader context, thereby offering a more comprehensive perspective on the overall significance of the research.]

Response1: [Thank you for the suggestions and the problems point out. According to the suggestions, we discussed the results and added the implications with a broader context in the section of “4. Discussion” on page 11-13. The revision were listed below:

This study analyzed the relationship between work-family conflict and turnover intention among rural kindergarten teachers, found the mediating effect of emotional exhaustion and the moderating effect of professional identity. With the development of society, more and more women enter the workplace, most of them are faced with the problem of balancing work and family, and kindergarten teachers are more prominent. This study revealed the influence mechanism of rural kindergarten teachers' work-family conflict on their turnover intention under the background of the new era, and provided the empirical basis for the professional development of rural kindergarten teachers and the practice of stabilizing kindergarten teachers in rural areas for the policymakers.

The results of the study also enlighten the kindergarten managers need to enhance the professional identity of rural kindergarten teachers in various ways, such as strengthening their professional identity education, improving the treatment of teachers, providing more humanistic care for teachers, and reducing their professional pressure, etc. The policymakers also should pay more attention to the negative effect brought by work-family conflict of rural kindergarten teachers. Some measures should be taken to ease their work-family conflict and emotional exhaustion, for example, provide convenience for rural kindergarten teachers to commute and reduce their time cost of commuting; increase the financial subsidies for kindergarten teachers in rural areas; provide specialized mental health training, etc. The teachers should also balance family life and work tasks well, and regulate their emotional state. ]

Round 2

Reviewer 2 Report

Comments and Suggestions for Authors
  • Your revisions have significantly improved the manuscript. Addressing the further suggestions provided will enhance the quality and impact of your research. Keep ensuring clarity, coherence, and adherence to the journal's guidelines in your final submission.

  • Ensure the literature review in the introduction critically engages with the cited studies, highlighting how your research builds on or diverges from existing work.
  • Clearly articulate the theoretical framework guiding your study.
  • Ensure a logical flow from general context to specific research questions to improve reader engagement.
  • Include a plagiarism report with the final submission to verify the similarity index.
  • Ensure all sources are correctly cited and emphasize original contributions and insights derived from your research.
  • Ensure the presentation of results is clear and concise. Double-check all tables and figures for accuracy and clarity.
  • Provide detailed explanations of the statistical analyses and their implications to enhance reader understanding.
  • Discuss the broader implications of your findings for theory, practice, and policy.
  • Consider potential limitations of your study and suggest areas for future research.
  • Ensure that all conclusions drawn are directly supported by the data presented in the results section.

Author Response

Dear Reviewer:

Thank you for your letter and for the reviewers’ comments concerning our manuscript entitled “Chinese Rural Kindergarten Teachers’ Work–Family Conflict and Their Turnover Intention: The Role of Emotional Exhaustion and Professional Identity”(ID: behavsci-3034573). Those comments are all valuable and very helpful for revising and improving our paper, as well as the important guiding significance to our researches. We have studied comments carefully and have made correction which we hope meet with approval. Revised portion are highlighted with yellow in the paper. The main corrections in the paper and the responds to the reviewer's comments are as flowing:

Comment1: [Ensure the literature review in the introduction critically engages with the cited studies, highlighting how your research builds on or diverges from existing work.]  

Response1: [Thank you for the suggestions, first, we rechecked the section of introduction to ensure the literature review critically engages with the cited studies; second, we added the content of how our research builds on or diverges from existing work. Please see the section of “the present study” on page 5.]

Comment2: [Clearly articulate the theoretical framework guiding your study.]

Response2: [Thank you for the suggestions, we have clearly introduced the relevant theories guiding the study, which includes Scarcity Hypothesis, JD-R model, and Risk-Buffering Hypothesis. These theories were described in the section of introduction (section1.1-section1.3)to support the hypothesis. In the section of “the present study”, we also stated the theoretical framework to further proved the hypothesis.]

Comment3: [Ensure a logical flow from general context to specific research questions to improve reader engagement.]

Response3: [Thank you for the suggestions, we checked and revised the whole article to ensure the logic of the text was from general context to specific research questions.]

Comment4: [Include a plagiarism report with the final submission to verify the similarity index.]

Response4: [Thank you for the suggestions, after finishing the revision, we used the software of “Turnitin” to test the similarity index again, the exclude matches was set 1%, the result indicated a 12% similarity index. We attached a plagiarism report to verify the similarity index.]

Comment5: [Ensure all sources are correctly cited and emphasize original contributions and insights derived from your research.]

Response5: [Thank you for the suggestions, first, we rechecked the whole article to ensure all sources are correctly cited; second, we also added the original contributions and insights of our own study in the end of the section of introduction on page5.]

Comment6: [Ensure the presentation of results is clear and concise. Double-check all tables and figures for accuracy and clarity.]

Response6: [Thank you for the suggestions, we rechecked the results to ensure the presentation is clear and concise, and the accuracy and clarity of all the tables and figures.]

Comment7: [Provide detailed explanations of the statistical analyses and their implications to enhance reader understanding.]

Response7:[Thank you for the suggestions. We provided the detailed explanations of the statistical analyses and their implications to enhance reader understanding. Please see the revision in the section of results on page 7-10.]

Comment8: [Discuss the broader implications of your findings for theory, practice, and policy.]

Response8: [Thank you for the suggestions. According to the suggestions, we added the broader implications of our findings for theory, practice, and policy in the section of “4.4. Implications and Limitations” on page 13-14.]

Comment9: [Consider potential limitations of your study and suggest areas for future research.]

Response9: [Thank you for the suggestions. We added the potential limitations of our study and suggest areas for future research in the section of “4.4. Implications and Limitations” on page 13-14.]

Comment10: [Ensure that all conclusions drawn are directly supported by the data presented in the results section.]

Response10: [Thank you for the suggestions. We rechecked the conclusions to ensure that all conclusions drawn are directly supported by the data presented in the results.]
